# Impact of the COVID-19 Pandemic on the Metabolic Control of Diabetic Patients in Diabetic Retinopathy and Its Screening

**DOI:** 10.3390/jcm11237121

**Published:** 2022-11-30

**Authors:** Pedro Romero-Aroca, Marc Baget-Bernaldiz, Ramon Sagarra, Esther Hervás, Reyes Blasco, Julia Molina, Empar F. Moreno, Eugeni Garcia-Curto

**Affiliations:** 1Ophthalmology Service, Hospital Universitario Sant Joan de Reus, 43204 Reus, Spain; 2Medicine and Surgery Departement, Medicine and Health Science Faculty, Universitat Rovira & Virgili, 43204 Reus, Spain; 3Pere Virgili Institute for Health Research (IISPV), 43204 Reus, Spain; 4Health Care Area Reus-Priorat, Institut Catala de la Salut, 43202 Reus, Spain

**Keywords:** telemedicine, diabetic retinopathy, diabetic macular edema, epidemiology, COVID-19

## Abstract

(1) Background: Diabetic retinopathy (DR) is a complication of diabetes mellitus (DM), screening programs of which have been affected by the COVID-19 pandemic. The aim of the present study was to determine the impact of the COVID-19 pandemic on the screening of diabetes patients in our healthcare area (HCA). (2) Methods: We carried out a retrospective study of patients with DM who had attended the DR screening program between January 2015 and June 2022. We studied attendance, DM metabolic control and DR incidence. (3) Results: Screening for DR decreased in the first few months of the pandemic. The incidence of mild and moderate DR remained stable throughout the study, and we observed little increase in severe DR, proliferative DR and neovascular glaucoma during 2021 and 2022. (4) Conclusions: The current study shows that during the COVID-19 pandemic, screening program attendance decreased during the year 2020, which then recovered in 2021. Regarding the most severe forms of DR, a slight increase in cases was observed, beginning in the year 2021. Nevertheless, we aimed to improve the telemedicine systems, since the conditions of a significant proportion of the studied patients worsened during the pandemic; these patients are likely those who were already poorly monitored.

## 1. Introduction

In December 2019, a group of pneumonia cases was identified by genomic sequencing as a novel coronavirus, SARS-CoV-2 [1], in the city of Wuhan (China). Due to the characteristics of this newly identified strain—namely its high communicability, spread and risk of death—the WHO declared COVID-19 to be a pandemic on 30 January 2020 [2]. The collapse of the health network, together with the complete lockdown of citizens, resulted in the suspension of all scheduled outpatient medical activity, along with the relevant examinations for the diagnosis and monitoring of certain pathologies.

Among such pathologies is diabetes mellitus (DM), a disease which has been steadily increasing worldwide for several years [3]. The latest report published by the International Diabetes Federation (IDF) estimates that 351.7 million people of active age (20–64 years) currently suffer with diagnosed or undiagnosed diabetes; this number is set to increase to an estimated 417.3 million by 2030 and 486.1 million by 2045 [4].

One of the complications of DM is diabetic retinopathy (DR), a form of microangiopathy of diabetes which targets small vessels—in this case, the vessels of the retina. The cause is high levels of glycemia, which induce deposition in the cells in the form of AGE proteins, inducing cell injury with a loss of pericytes and endothelial cells, as well as an increase in oxidative stress with subsequent cell injury. DR is one of the main causes of blindness and preventable visual impairment in Europe today, especially when it develops into diabetic macular edema (DME) or proliferative DR (PDR). The harmful effects of DM on vision have a significant impact on health spending and require screening programs that allow regular monitoring, early detection and timely treatment [5]. The COVID-19 pandemic has impacted the screening of DM patients, which depends on the fidelity of the patients to the screening system.

The aim of the present study was to determine the impact of COVID-19 on screening programs for diabetes patients at our non-mydriatic fundus camera units at Hospital Universitario, Sant Joan de Reus.

## 2. Materials and Methods

### 2.1. Study Design

We designed a retrospective study of patients with DM who had attended the DR screening units between January 2015 and June 2022 at Hospital Universitari Sant Joan de Reus, whose reference population is 200,318 inhabitants. Based on WHO estimates, the total number of DM patients in our health area is around 17,792. The present study included 16,152 patients with DM who had attended our DR screening program during the study period: 98 patients had type 1 DM (T1DM), and 16054 had type 2 DM (T2DM).

### 2.2. Ethical Adherence

This study adhered to the legal requirements of our local ethics committee (approval CEIM 028/2018), in accordance with the revised guidelines of the Declaration of Helsinki. Informed consent was obtained from all participants in the study.

### 2.3. Inclusion Criteria

The inclusion criteria covered any patient with T1DM and T2DM overseen by the family doctors in our HCAs.

### 2.4. Exclusion Criteria

The exclusion criteria were as follows: patients included in group III of diabetes, along with other specific types (i.e., diseases of the exocrine pancreas, endocrinopathies, genetic defects of ß-cell function, genetic defects in insulin action), patients included in group IV of diabetes and patients presenting with gestational diabetes mellitus (GDM).

### 2.5. Objectives

The objectives of the study were as follows: to assess the impact of COVID-19 on the metabolic control of DM patients; to record the attendance numbers for DR screening during the period of the pandemic and determine the levels of glycemia, HbA1c, systolic and diastolic blood pressure and degree of renal involvement through the value of the glomerular filtration rate; to record the incidence of patients with DR detected at our non-mydriatic camera units; and to determine the flow of patients with DR, classified according to the International DR guidelines.

### 2.6. Methods

This study was carried out using an initial retinograph set at 45°, centered between the macula and the temporal side of the papilla. A diagnosis of DR was then performed using three 45° retinographs, according to the Joslin Vision Network (one centered on the macula, a second centered on the nasal side of papilla and a third centered at the temporal superior) using a Topcon NW400 retinal camera. The circuit and the technique are described in more detail elsewhere [6]. DR was diagnosed when microaneurysms were present in the fundus retinograph.

The opacity of the media makes it difficult to see the fundus, especially if retinography is performed using a non-mydriatic camera; however, images taken using the new equipment have noticeably improved. Around 3% of patients, on the other hand, were not able to visualize the image clearly. In these cases, we visited the patients in their respective ophthalmology services in person, even during the pandemic, and resorted to other diagnostic systems such as biomicroscopy, optical coherence tomgraphy (OCT) and angio-OCT.

DR severity was classified according to the International Council of Ophthalmology (ICO) [7] as (i) mild DR with microaneurysms only, (ii) moderate DR (microaneurysms, hard exudates, hemorrhages and venous abnormalities), (iii) severe DR (the above, together with one of the following: >20 hemorrhages in each quadrant, venous anomalies defined in 2 quadrants, intra-retinal microvascular abnormalities in 1 quadrant, no signs of proliferation, and (iv) proliferative DR, defined as the presence of neovascularization. Neovascular glaucoma (NVG) is a severe form of secondary glaucoma characterized by the proliferation of fibrovascular tissue in the anterior chamber angle.

### 2.7. Studied Risk Factors

The epidemiological risk factors included in the study were age and sex, duration of diabetes mellitus, arterial hypertension (indicated by a systolic/diastolic (normal value = 140/90 mm Hg) measurement according to the report of the sixth joint National Committee on the Prevention, Detection, Evaluation and Treatment of High Blood Pressure) and whether the patient is taking antihypertensive medications. Glycosylated hemoglobin (HbA1c) levels were defined according to recommendations made by the American Diabetes Association [8,9,10], and the estimated glomerular filtration rate (eGFR), as measured by the chronic kidney disease epidemiology collaboration equation CKD-EPI, was estimated on urine collection.

### 2.8. Statistical Analysis

Data were analyzed using SPSS, version 22.0 (IBM^®^ Statistics, Chicago, IL, USA). The specific statistical study carried out and the specific type of test applied depended on the data obtained and their distribution.

For descriptive statistics, the mean, standard deviation, and the 95% confidence interval for the mean, the median and the maximum and minimum values were used for quantitative variables. For qualitative variables, the absolute and relative frequencies and the percentages of each category were used.

The differences between two means were calculated by Student’s *t*-test for independent samples (normal variable). Differences were considered statistically significant at *p* < 0.05. Comparisons of more than two means were carried out through the analysis of variance (ANOVA for normal variables).

## 3. Results

### 3.1. Evolution of the SARS-CoV-2 Pandemic

Firstly, we wanted to reflect on the impact that the SARS-CoV-2 pandemic had on attendance at DR screening.

Figure 1 shows the evolution of the COVID-19 pandemic in our HCA from January 2020 to December 2021 (black line). It shows an exponential increase in cases in March 2020. However, it is necessary to take into account the poor diagnostic capacity of the first wave. As reported a posteriori, the real incidence was estimated to be 10 times higher. From March to May 2020, because of the restrictions and confinement measures established by the State of Emergency, the number of incidences decreased. Subsequently, a second wave was reported in November of the same year. The reduction in this wave was slower than the first wave and overlapped the third wave at the beginning of January 2021. The accumulated incidence was greater than in the first wave because of a better diagnostic capacity; however, this was due to better hospital care, leading to improved rates of survival and better control of cases.

The last very high peak in the incidence of COVID-19, during January–February 2022, was due to the Omicron variant. Figure 1 shows, however, that this did not particularly impact DR screening. The latest data we have are from June 2022, as the other health authorities were no longer reporting incidences.

In the eight-year period from 1 January 2015 to 31 December 2021, we screened a total of 16,152 patients with DM (98 T1DM and 16054 T2DM). Each patient visited the screening units 2.38 times on average during the seven-year follow-up period.

The data in Figure 1 and Table 1 show a reduction in the total number of patients screened in 2020, followed by an increase in the numbers in 2021, even higher than in the years prior to the pandemic. During the study period, there were never more than 6000 patients screened in any given year until the peak in 2021. The total number of DM patients screened was 16152, of which 8621 were male (53.38%) and 7530 were female (46.62%). The mean age was 67.93 ± 8.69 (12–89) years and the mean DM duration was 9.09 ± 6.79 (1–54) years.

Table 1 shows the screening levels from 2015 to 2021. During 2020, attendance at the screening units reduced and there was a significant increase in the following year, 2021. The greater number of males screened reflects the proportion of DM in the population. The statistical analyses showed no significant differences in the variables of sex, mean age, type or duration of DM.

### 3.2. Demographic Data of DR Patients

The total number of screened patients with DM was 16152, of which 8621 were males (53.38%) and 7530 were females (46.62%). The mean age was 67.93 ± 8.69 (12–89) years and the mean DM duration was 9.09 ± 6.79 (1–54) years. According to the type of DM, 98 patients had type 1 DM and 16054 had type 2 DM.

The sample included a greater prevalence of men, which is in line with the prevalence of diabetes in the population; there were no differences with respect to sex prevalence throughout the study. Moreover, there were also no significant differences according to the mean age. No differences in the type of diabetes mellitus were significant throughout the study. The DM mean duration was similar throughout the follow-up. The differences in these four variables were not significant in statistical analysis.

Table 2 shows that only the changes in the mean values of HbA1c were significant. The average HbA1c between 2015 and 2019 was 7.45 ± 2.42%, compared with 7.51 ± 1.32% in 2020, 7.52 ± 1.68% in 2021 and 7.54 ± 1.37% in the first six months of 2022. Differences were measured by Student’s *t*-test at a significance of *p* < 0.001.

On the other hand, there were no statistically significant differences in age, sex, DM duration and mean glomerular filtration rate values; the mean values for systolic and diastolic blood pressure showed similar results.

Attending the number of tests per patient and per year, there was a mean of 1.18 ± 0.28 (1.15 to 1.22) in 2015–2019 against 0.96 ± 0.19 (0.79 to 1.18) in 2020–2022. A significant proportion of patients who should have attended screening in 2020 did attend in 2021. In total, 1097 patients invited to attend in 2020 were seen in 2021; that is, 17.97% of those seen in 2021 should have been seen in 2020. We have analyzed the data of these patients, and they did not show a significant difference in terms of risk factors, age (*p* = 0.502), sex (*p* = 0.374), DM duration (*p* = 0.324), HbA1c (*p* = 0.103), eGFR (*p* = 0.230), systolic arterial pressure (*p* = 0.330) and diastolic arterial pressure (*p* = 0.304).

### 3.3. Diabetic Retinopathy Study

Table 3 shows the incidence of DR in the screened patients. From 2015 to 2019, 5676.40 ± 439.75 patients (minimum 4911, maximum 5996) were screened on average. This number dropped to 3286 in 2020, and increased to 6804 in 2021, higher than the averages from previous years. During the first half of 2022, 2957 patients were screened, which suggests that 5914 patients will be screened this year, a figure similar to that prior to the pandemic.

Table 4 shows the results of the study on DR classification. From 2015 to 2019, the mean number with mild DR was 131.80 ± 21.61 patients, which decreased to 81 patients in 2020, then increased to 135 in 2021; after the first six months of 2022, 64 patients had mild DR. For moderate DR, the mean was 56.20 ± 15.7 affected patients, which increased to 62 in 2020, and to 66 in 2021. After the first six months of 2022, 32 patients were found to have mild DR, higher than the average of previous years; however, the results are not significant at *p* = 0.023. The numbers for severe DR were similar, with an average of 35 ± 5.29 patients between 2015 and 2019, a decrease to 36 in 2020 and an increase to 69 in 2021. In the first six months of 2022, 26 patients were affected, which is also significant at *p* = 0.041. For proliferative DR, the average number of patients affected in the period from 2015 to 2019 was 13.80 ± 3.60, increasing slightly to 15 patients in 2020 and, more significantly, to 29 in 2021. With 17 affected patients in the first six months of 2022, these changes are significant at *p* = 0.037.

Regarding the presence of diabetic macular edema (DME), the mean value from 2015 to 2019 was 45 ± 3.60 patients, falling to 19 cases in 2021 and increasing to 45 in 2021, with 23 patients affected in the first six months of 2022. These differences are not significant at *p* = 0.07.

Finally, for neovascular glaucoma secondary to DR, an average of 5 ± 1 patients were affected from 2015 to 2019; five patients were affected in 2020, which increased to seven patients in 2021, with three patients affected in the first six months of 2022. The differences between them were significant at *p* = 0.04; however, due to the small sample size, this should not be considered representative of any possible changes due to the pandemic.

## 4. Discussion

The present study focused on the changes in DR screening during the COVID-19 pandemic in our HCA. The study shows a reduction in the number of patients attending screening throughout 2020 (Table 3). However, Figure 1 shows that fewer patients attended during March and April 2020, and the numbers subsequently reduced slightly each time there was a new wave of COVID-19. Screening numbers recovered in 2021 and again in the first half of 2022. In 2021, we observed that 17.97% were patients who had been scheduled in 2020 but did not attend their appointment; however, a greater incidence of DR was not observed in these patients.

Overall, we have observed throughout this study that the COVID-19 pandemic has not generated a greater number of patients with DR; however, we did observe an increase in the number of patients with severe DR and proliferative DR during the years 2021 and 2022. In a small number of patients during the pandemic, who previously exhibited inferior metabolic control, more serious forms of poor metabolic control may have developed, which we were not able to detect in time.

In relation to the metabolic control of the patients, we observed worse levels in the 2015–2019 group compared with the 2020–2022 period; as seen in the results, the mean levels of HbA1c were 7.45 ± 2.42% in the first period (2015–2019) against increases in the period from 2020 to 2022 (HbA1c = 7.51 ± 1.32% in 2020, 7.52 ± 1.68% in 2021), decreasing again in the first six months of 2022 to values of 7.47 ± 1.37%. These global data from all of the patients examined can give us some idea of what has happened in patients with DM during the pandemic, with worse control during the years 2020 and 2021 that is now recovering in the year 2022. These data have been collected on a global scale; individually, however, there may still be large variations.

A different approach concerns the number of control tests that patients with DM have carried out during the pandemic. In the current study, we determined the amount of HbA1c data that we obtained from the sample. Thus, the study shows a range from 1.18 ± 0.28 (1.15 to 1.22) per patient extractions between 2015 and 2019, to 0.96 ± 0.19 (0.79 to 1.18) per patient between 2020 to 2022. Only in the first six months of 2022 did they reach a number of HbA1c tests similar to those prior to COVID-19. It is possible that the extractions were only performed in those patients with poorer glycemic control. Unfortunately, however, we do not have access to these data.

Regarding DR type, the study shows an increase in the crude data of screened patients with severe DR and proliferative DR in 2021; however, in terms of percentage, the increase is not as important. We may suppose that patients with poorer DM control worsened during the pandemic and went on to develop severe DR and proliferative DR.

With respect to diabetic macular edema, there was no increase in the number of patients. Pre-pandemic, there were 40–48 screened patients, reducing to only 19 patients in 2020 but recovering to pre-pandemic levels in 2021 with 42 patients, and 23 patients in the first six months of 2022. It is possible that diabetic macular edema requires other risk factors that have not been affected as much during the pandemic, such as the control of renal function. In this study, we observed that the changes in eGFR were not significant.

There are no data in the current literature on the impact of COVID-19 on the screening of patients with DM. There have indeed been publications on how to react during any future pandemic, following initial guidelines by the Royal College of Ophthalmology [11], who suggested that postponing DR screening might be necessary when facing severe staff shortages, a lack of personal protective equipment and an escalation of infection numbers. At the beginning of July 2020, Shih et al. [12] reported that the default rate was especially high in February 2020, when there was widespread fear of contacting COVID-19 among patients, although they reported a slight improvement in March 2020. As of 12 April 2020, we had no reported cases of COVID-19 infection among our own clinical staff, nor among any patients who had undergone DR screening.

Moreover, there have been publications on the levels of reduction in the number of intravitreal injections in the treatment of patients with DME due to the pandemic [13], which lead to vision loss in some patients who have not recovered their sight [14,15,16]. Furthermore, there have been numerous articles published on the importance of telemedicine in the treatment of patients with DM. Such systems can detect the presence of DR as early as possible, especially for DME or severe or proliferative DR [17,18,19,20,21]. Similarly, uses of new AI technologies applied to DR diagnosis are being developed, driven by the pandemic [22,23].

Regarding how the pandemic affected patients with DM, we do not know whether other complications have changed during the pandemic period, and we do not know the number of new patients; we only have data on the decrease in attending the screening programs of DR, and we only have the number of HbA1c per patient during the COVID-19 period, which decreased. Therefore, we can deduce that fewer tests were carried out per patient, which resulted in notable difficulties for the metabolic control of these patients.

Although the effect of vaccination was not the aim of the present study, we can assume that vaccination was the cause of higher patient attendance at screening consultations, as they had greater health security. The present study shows that having a telemedicine system strongly implanted in a healthcare area can reduce the effect of a novel pandemic that makes it difficult for patients to be physically present during patient medical visits. In any case, we must improve telemedicine systems, since a significant percentage of the patients studied worsened during the pandemic, most likely those who were already poorly treated.

The first limitation of this study is that it was carried out in our own HCA, which is an area with great sensitivity towards DR screening both for patients and primary care physicians, as it was one of the first areas ever to implement a screening program (in 2007) [24,25]. It is, therefore, difficult to extrapolate our results to other areas of Spain, given the significant differences in the implementation of screening programs. Moreover, the study was based only on diabetic patients who were already diagnosed and who were known by the health system. We do not know what has happened to patients who were not diagnosed with diabetes during the COVID-19 pandemic. Finally, we described changes in the screening program, not the effect of the pandemic in DR and DME treatment. Patients described in the study were naïve and detected during screening; these patients were referred to the retina section of the hospital to be treated. The objective of this study was not to determine the effect of the pandemic on the treatment of DR.

## 5. Conclusions

This study shows that during the COVID-19 pandemic, attendance to the screening program decreased during the year 2020 and then recovered in 2021. Regarding the most severe forms of DR, a slight increase was observed from the year 2021. In summary, we should improve telemedicine systems, since a significant percentage of the patients studied worsened during the pandemic, which resulted in clear difficulties in the metabolic control of these patients.

## Figures and Tables

**Figure 1 jcm-11-07121-f001:**
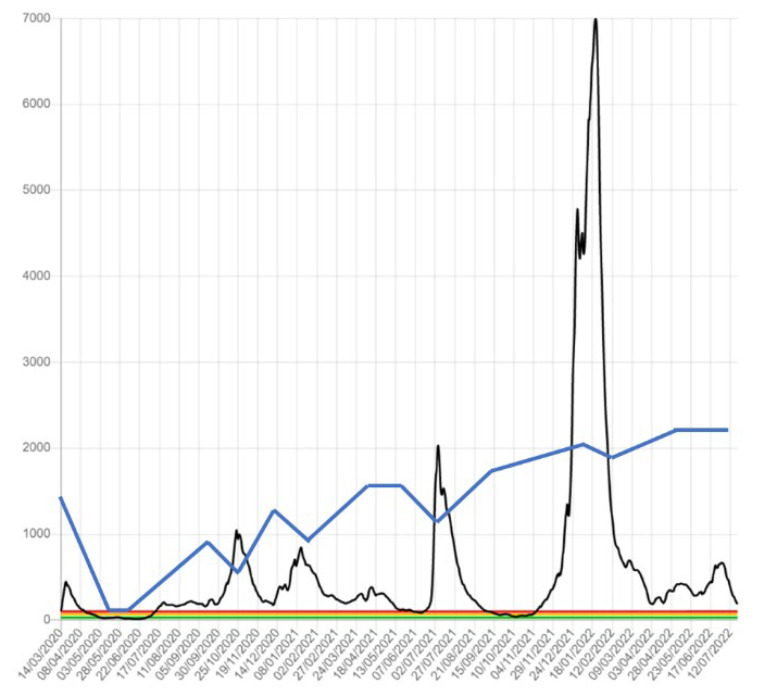
Weekly confirmed cases of COVID-19 in our healthcare area in Catalonia (Spain) from March 2020 to June 2022; the image also shows both the percentage of patients with COVID-19 and attendance at DR screening. Blue line = weekly confirmed cases of COVID-19 in our HCA from 1 March 2020 to 30 June 2022. Black line = weekly attendance at DR screening from 1 March 2020 to 30 June 2022 in our HCA.

**Table 1 jcm-11-07121-t001:** DR screening results, number of patients screened and percentage with respect to the total number of patients with DM in the study.

Year	Screened Patients	Percentage (%)
2015	5996	37.12
2016	5801	35.91
2017	4911	30.40
2018	5933	36.73
2019	5741	35.54
2020	3286	20.34
2021	6104	37.79
2022 *	2957 *	19.30 *

* Data from January to June 2022.

**Table 2 jcm-11-07121-t002:** HbA1c, arterial systolic and diastolic tension and eGFR estimated glomerular filtration rates along the seven-year follow-up study.

	2015	2016	2017	2018	2019	2020	2021	2022	*p*
Age	66.87 ± 8.57	67.02 ± 7.96	68.04 ± 9.12	67.12 ± 8.56	68.19 ± 8.77	66.97 ± 7.79	67.55 ± 8.02	68.01 ± 8.34	0.502
Sex (women %)	46.71	46.55	47.12	46.71	47.01	45.52	46.37	46.87	0.374
DM duration in years	9.11 ± 6.57	9.01 ± 6.78	9.12 ± 6.81	9.07 ± 6.85	9.13 ± 6.72	9.02 ± 6.78	9.15 ± 6.91	9.03 ± 6.98	0.324
HbA1c (%)	7.43 ± 3.09	7.47 ± 2.80	7.43 ± 2.57	7.46 ± 1.92	7.49 ± 1.72	7.51 ± 1.32	7.52 ± 1.68	7.47 ± 1.37	<0.001
eGFR (ml/min/1.73 m^2^)	80.33 ± 23.56	81.29 ± 22.47	79.33 ± 24.33	80.23 ± 21.34	79.29 ± 23.56	81.36 ± 20.21	80.96 ± 24.81	80.66 ± 24.52	0.071
Annual number of analytics per patient	1.21	1.15	1.22	1.17	1.18	0.79	0.91	1.18	
Systolic arterial pressure (mmHg)	137.10 ± 12.90	135.97 ± 15.29	134.20 ± 20.11	136.20 ±18.3	136.11 ± 19.89	135.47 ± 20.53	134.50 ± 23.33	134.50 ± 21.67	0.134
Diastolic arterial pressure (mmHg)	79.97 ± 7.54	76.74 ± 7.20	78.40 ± 6.97	78.67 ± 7.27	79.18 ± 10.65	78.63 ± 11.48	77.92 ± 11.22	78.49 ± 11.21	0.087

**Table 3 jcm-11-07121-t003:** Incidence of DR each year in the screened patients.

Year	Screened Patients	Diabetic Retinopathy Patients	Percentage of Screened DR Patients %
2015	5996	220	3.68
2016	5801	208	3.58
2017	4911	213	4.33
2018	5933	277	4.66
2019	5741	266	4.61
2020	3286	194	5.91
2021	6104	299	4.89
2022 *	2957	139	4.70

* data only for 6 months.

**Table 4 jcm-11-07121-t004:** Classification of DR and its complications.

		2015	2016	2017	2018	2019	2020	2021	2022 *	*p*
DR classification	Mild	130 * (2.16%) **	116 (1.99%)	110 (2.23%)	165 (2.78%)	138 (2.40%)	81 (2.46%)	135 (2.21%)	64 (2.16%)	0.017
	Moderate	47 (0.08%)	42 (0.72%)	49 (0.99%)	62 (1.04%)	81 (1.41%)	62 (1.88%)	66 (1.08%)	32 (1.08%)	0.023
	Severe	28 (0.46%)	37 (0.63%)	42 (0.85%)	36 (0.60%)	32 (0.55%)	36 (1.09%)	69 (1.13%)	26 (0.87%)	0.041
	Proliferative	15 (0.25%)	13 (0.22%)	12 (0.24%)	14 (0.23%)	15 (0.26%)	15 (0.45%)	29 (0.47%)	17 (0.57%)	0.037
DR complications	Diabetic macular edema	48 (0.8%)	49 (0.84%)	45 (0.91%)	44 (0.74%)	40 (0.69%)	19 (0.57%)	45 (0.73%)	23 (0.77%)	0.070
	Neovascular glaucoma	6 (0.10%)	5 (0.08%)	6 (0.12%)	4 (0.06%)	4 (0.06%)	5 (0.15%)	7 (0.11%)	3 (0.10%)	0.040

* Number of patients, ** percentage of patients with DR type versus number of patients screened this year.

## Data Availability

Not applicable.

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
