# Peer review of "Impact of the COVID-19 Pandemic on the Metabolic Control of Diabetic Patients in Diabetic Retinopathy and Its Screening"

_jcm, 2022, doi:10.3390/jcm11237121_

Round 1
Reviewer 1 Report (Previous Reviewer 1)
1. What will be the impact of this study in terms of future implications? Please state in the abstract itself.
2. What about the medication status? Did the authors check the patients if were taking some medicines during the viral infection? What was the impact of medications on DM or DR?
3. What about vaccination status and its impact on this study?
4. The conclusions section needs to be improved. Please include the future implications of this study.
5. What is the mechanistic relation between diabetes and diabetic retinopathy? Please discuss.
6. Please include a separate section stating the limitations of the present study. This is a must for such types of studies.
Author Response
Responses to reviewer 1
- What will be the impact of this study in terms of future implications? Please state in the abstract itself.
Thank you for your comment, we have included the following sentence in the conclusions
“In any case, we have to improve telemedicine systems since a not insignificant part of the patients studied worsened during the pandemic, surely those who were already poorly controlled.”
- What about the medication status? Did the authors check the patients if were taking some medicines during the viral infection? What was the impact of medications on DM or DR?
We do not know the impact of possible medications to treat COVID-19, since it was not the subject of the study.
None of the patients included in the study were hospitalized for COVID-19 or suffered from severe COVID-19 disease. We have included this paragraph in results.
- What about vaccination status and its impact on this study?
As the effect of vaccination is not the aim of the study, we have not included this data in the study. In any case, we can assume that vaccination was the cause of greater patient attendance at screening consultations, due to greater health security. We have included this paragraph in the discussion section.
- The conclusions section needs to be improved. Please include the future implications of this study.
We changed conclusions in abstract and in text as follow,
“The current study shows that during the COVID-19 pandemic attendance to the screening program decreased during the year 220, which recovered in 2021. Regarding the most severe forms of DR, a slight increase was observed from the year 2021. In any case, we have to im-prove telemedicine systems since a not insignificant part of the patients studied worsened during the pandemic, surely those who were already poorly controlled.”
- What is the mechanistic relation between diabetes and diabetic retinopathy? Please discuss.
Thanks for your commentary , we changed the introduction as follows,
“One of the complications of DM is diabetic retinopathy (DR), that is one of the forms of microangiopathy of diabetes, that is, the involvement of the small vessels, in this case the vessels of the retina. The cause is the elevated levels of glycemia that induce their deposition in the cells in the form of AGE-proteins, inducing cell injury with loss of pericytes and endothelial cells, as well as an increase in oxidative stress with subsequent cell injury. DR is one of the main causes of blindness and preventable visual impairment in Europe today, especially when it develops to diabetic macular edema (DME) or proliferative-DR (PDR). These harmful effects of DM on vision have a significant impact on health spending and require screening programmes that allow regular monitoring, early detection, and timely treatment [5].”
- Please include a separate section stating the limitations of the present study. This is a must for such types of studies.
Dear reviewer, we include limitations at the end of the discussion (not as a separate section, because the text model does not include this section)
We include following limitations,
Limitations of the study include, first it was carried out in our own HCA, which is an area with great sensitivity towards DR screening both by patients and primary care physi-cians, as it was one of the first areas ever to implement a screening programme, in 2007 [24, 25], it is therefore difficult to extrapolate our results to other areas of Spain, given the significant differences in the implementation of screening programmes in other health areas. Also, the study was based only on diabetic patients who were already diagnosed and who were known by the health system. We do not know what has happened in patients who were not diagnosed with diabetes during the covid period. Finally, we describe changes in screening program not the effect of pandemia in DR and DME treatment, patients described in the study are naïve and detected in screening, these patients were referred to the retina section of the hospital to be treated, and the objective of this study was not to determine the effect of the pandemic on the treatment of DR.

Reviewer 2 Report (Previous Reviewer 2)
Thank you for your point-to-point responses. Two more small comments:
1. In line 316, “…, which actually decreased and …” maybe “increased”?
2. The author may still need to enhance the expression of “significance of current study” in discussion section. Actually, the current study was based on “screening ”, instead of “diagnosing” (because the diagnosing of diabetic retinopathy should base on a combination of FA/OCTA, OCT and fundus image). The author mentioned that the core meaning of present study was “raising the importance of telemedicine system”, but the online diabetic retinopathy screen system had been developed by Google and many other researchers, some of them has even been distributed. The author still needs to mining core influences of COVID19 in managing diabetic patients.
Author Response
Responses to reviewer 2
- In line 316, “…, which actually decreased and …” maybe “increased”?
Dear reviewer, thanks for your commentary. We changed conclusions as follows,
“The current study shows that during the COVID-19 pandemic attendance to the screening program decreased during the year 220, which recovered in 2021. Regarding the most severe forms of DR, a slight increase was seen from the year 2021. In any case, we have to im-prove telemedicine systems since a not insignificant part of the patients studied worsened during the pandemic, surely those who were already poorly controlled.”
- The author may still need to enhance the expression of “significance of current study” in discussion section. Actually, the current study was based on “screening ”, instead of “diagnosing” (because the diagnosing of diabetic retinopathy should base on a combination of FA/OCTA, OCT and fundus image). The author mentioned that the core meaning of present study was “raising the importance of telemedicine system”, but the online diabetic retinopathy screen system had been developed by Google and many other researchers, some of them has even been distributed. The author still needs to mining core influences of COVID19 in managing diabetic patients.
Dear reviewer, I have changed the expression diagnosis to screening in the discussion section. Likewise, it is true that there are diagnostic systems by reading retinal images, but these systems, unfortunately, have not yet been accepted by the health screening programs of the national health systems, so we can only hope that in the future These systems help us to better screen diabetic retinopathy.

Round 2
Reviewer 1 Report (Previous Reviewer 1)
The authors successfully responded to the reviewer's comments and updated the manuscript as well.
This manuscript is a resubmission of an earlier submission. The following is a list of the peer review reports and author responses from that submission.
Round 1
Reviewer 1 Report
1. What will be the future impact of this study? How this study could help the scientific community? Please highlight these details.
2. The study is just providing the data of screened patients during the COVID-19 period but what if the number of patients was increasing? The study doesn’t answer or focus on key questing or doesn’t conclude any future aspect of the study.
Reviewer 2 Report
The author tried to use follow-up data to explore the influences of COVID-19 on diabetic retinopathy, which is potentially meaningful in managing DM patients. Here are my comments:
1. It is generally known that diabetic retinopathy is highly associated with the duration of diabetes history and general control of glycemia when each follow-up. As the author only mentioned glycemia, the relationship among diabetes / DM / COVID-19 influence were somehow less convincing. Adding relative information and taking diabetic history into account is highly recommended.
2. As the mean age of the database was over 60y/o, qualities of fundus images may affect by cataracts, which small microaneurysms may be missed. So in the general ophthalmology clinic, the screening of DM is mainly based on fundus fluorescence for patients with 50y/o who suffered from diabetes for years, if possible. Fundus fluorescence is a key step in managing diabetic patients, the author should mention this information in the limitation section if lack of these data.
3. The information shown in Figure 2 seems to have covered Figure 1. Please refrain to show redundant figures.
4. According to the results shown in Table 1, the patients' number of 2021 showed a bounce compared to 2020. One may think that data in 2020 may be “those patients who have to follow-up despite of COVID-19 pandemic”, these patients may be potential severer DM. Furthermore, the 2021 data may somehow be like a data-delay condition, instead of those patients whose DM conditions got severed. Because the author did not mention how many patients in 2021 overlapped those follow-upped in 2020 (patients visited the hospital both in 2020 and 2021), how did the author consider this point?
5. The discussion part should be reorganized, instead of repeating the results section. The authors should deeply insight into the data and discuss the potential social factors of showed data, for example, increased DM prevalence affected by lack of medicine due to COVID-19? Or longer stay-at-home time? Or any other reasons. Analyzing these aspects may indicate us to manage DM patients in the future.
6. In addition to comment #5, the conclusion section seems also did not summarize the core content or have feasible indications of the clinical work, it should be reorganized.